# Impact of Gestational Diabetes and Hypertension Disorders of Pregnancy on Neonatal Outcomes in Twin Pregnancies Based on Chorionicity

**DOI:** 10.3390/jcm12031096

**Published:** 2023-01-31

**Authors:** Yi Liu, Dayan Li, Yang Wang, Hongbo Qi, Li Wen

**Affiliations:** 1Department of Obstetrics, The First Affiliated Hospital of Chongqing Medical University, Chongqing 400016, China; 2Women and Children’s Hospital of Chongqing Medical University, Chongqing 400014, China; 3State Key Laboratory of Maternal and Fetal Medicine of Chongqing Municipality, The First Affiliated Hospital of Chongqing Medical University, Chongqing 400016, China

**Keywords:** gestational diabetes, hypertension disorders of pregnancy, twin pregnancies, chorionicity, intertwin birthweight discordance, small for gestational age

## Abstract

Objectives: The objective of this study was to investigate the impact of the co-existence of gestational diabetes (GDM) and hypertension disorders of pregnancy (HDP) on neonatal outcomes in twin pregnancies based on chorionicity. Methods: A retrospective study of 1398 women with twin pregnancies was performed between January 2016 and December 2021. The effects of GDM and HDP on neonatal outcomes were assessed by logistic regression models. An additional stratified analysis was conducted to estimate the effects based on chorionicity (monochorionic (MC) and dichorionic (DC)). Results: The incidence of the co-existence of GDM and HDP was 3.8%. The presence of GDM increased the likelihood of HDP only in women with MC twin pregnancies (OR, 2.13; 95% CI 1.08–4.19). After adjustments, co-existence of GDM and HDP was positively associated with gestational age (β, 1.06; 95% CI 0.43–1.69) and birthweight (β, 174.90; 95% CI 8.91–340.89) in MC twin pregnancies, while no associations were found between co-existence of GDM and HDP and neonatal outcomes in DC twin pregnancies. However, HDP was negatively associated with birthweight (β, −156.97; 95% CI (−257.92, −56.02)) and positively associated with small-for-gestational-age (SGA) (OR, 2.03; 95% CI 1.02–4.03) and discordant twins (OR, 2.83; 95% CI 1.78–4.48) in DC twin pregnant women without GDM. Conclusions: Our results suggested that GDM leads to an increased risk of HDP only in MC twin pregnancies, but GDM seemed to attenuate the adverse effects of HDP on perinatal outcomes in both MC and DC twin pregnancies. Further investigation is needed to explain these intriguing findings.

## 1. Introduction

With the development of assisted reproductive technology (ART) and delayed child-bearing, the prevalence of twin pregnancies has been increasing worldwide over the past 5 decades [1]. Compared to singleton pregnancy, twin pregnancies are deemed high-risk in terms of increased perinatal morbidity and mortality due to preterm birth (PTB), low birthweight, small for gestational age (SGA), intrauterine growth restriction (IUGR), admission to the neonatal intensive care unit (NICU) and complications of pregnancy [2]. 

Gestational diabetes mellitus (GDM) and hypertension disorders of pregnancy (HDP) are two of the most common pregnancy complications. In recent studies based on different diagnostic criteria and populations, researchers found that the prevalence of GDM in twin pregnancies varied from 8.48% to 23.9% [3,4,5], which is much higher than that in singleton pregnancy. Moreover, in a large retrospective study reported by Hiersch et al., HDP including gestational hypertension (GH) and preeclampsia (PE) was more frequent among mothers with singleton pregnancy affected by GDM, but this finding was not found in twins [6]. Therefore, more evidence on the association between GDM and HDP in twin pregnancies is needed.

In addition, GDM has been associated with macrosomia or large for gestational age (LGA) in singleton neonates [7], but conflicting results regarding the association between GDM and LGA in twin pregnancies have been reported [6,8,9,10,11]. Conversely, GDM has been reported to decrease the rate of SGA or to be associated with a lower risk for SGA in twin pregnancies [9,12,13,14]. In our previous study, the rate of SGA in twin pregnant women with GDM was lower than that in women without GDM [15]. However, HDP has been reported to be associated with SGA in both singleton and twin pregnancies [16,17]. It appears the effects of GDM and HDP are contradictory in twin pregnancies. Growth discordance between twins is a specific but common issue in twin pregnancies; namely, one fetus reaches the appropriate size while the co-twin is SGA, or the intertwin birthweight difference is greater than 20% [18]. Although associations between HDP and SGA as well as intertwin birthweight discordance in twin pregnancies have been previously reported [19,20,21], the effects of these two simultaneously occurring factors on neonatal outcomes in twin pregnancies are still unclear. 

Our first aim was to investigate the association of GDM with HDP, and our second aim was to assess the combined effect of GDM and HDP on neonatal outcomes in twin pregnancies based on chorionicity.

## 2. Materials and Methods

### 2.1. Study Population

In the present study, we prospectively collected the data of a retrospective cohort study of twin pregnancies. A total of 2006 women with twin pregnancies were hospitalized at the First Affiliated Hospital of Chongqing Medical University between January 2016 and December 2021. Twin pregnancies were identified by searching the hospital information system (HIS). The inclusion criteria were as follows: accurate chorionicity, gave birth at our institution at delivery ≥ 28 weeks gestation. We excluded women with twin pregnancies and pre-existing diabetes and hypertension; women without oral glucose tolerance test (OGTT) results; women who experienced intrauterine death of one or both fetuses; women with twin-to-twin transfusion syndrome (TTTS), twin anemia-polycythemia sequence (TAPS) and twin reverse artery perfusion syndrome (TRAPS); and women with incomplete electronic medical records. The study received approval from the Ethics Committee of the First Affiliated Hospital of Chongqing Medical University (No. 201530).

### 2.2. Study Variables and Outcomes

Maternal sociodemographic data on prepregnancy age, height, weight, body mass index (BMI), gravidity, parity and mode of conception (assisted reproductive technology (ART) use or spontaneous conception) were collected. Chorionicity (monochorionic or dichorionic) was identified during 11 to 14 gestational weeks according to the method described in a previous study or checked after delivery [15]. Gestational weight gain (GWG) was obtained via the maternal weight prior to delivery minus the prepregnancy weight. Gestational age (GA) was determined based on the larger fetus’s crown–rump length (before 14 weeks gestation) or head circumference (after 14 weeks gestation) in cases of spontaneous conception, and based on the timing of in vitro fertilization for mothers who conceived with the aid of assisted reproductive technology (ART). We defined preterm birth (PTB) as delivery prior to 34 gestational weeks, due to the mean gestational age for twin pregnancies being approximately 36 gestational weeks in our center. Delivery modes included vaginal delivery and cesarean delivery.

GDM was diagnosed when the conditions of fasting glucose ≥ 5.1 mmol/L, and/or 1 h glucose ≥ 10.0 mmol/L, and/or 2 h glucose ≥ 8.5 mmol/L during 23 to 28 gestational weeks according to the IADPSG criteria were recorded via a 75 g oral glucose tolerance test (OGTT). HDP included gestational hypertension (defined as systolic blood pressure ≥ 140 mmHg and/or diastolic blood pressure ≥ 90 mmHg after 20 weeks of gestation), preeclampsia (PE) (defined as the onset of hypertension ≥ 140/90 mmHg and proteinuria after 20 weeks of gestation) and eclampsia (characterized by the occurrence of 1 or more generalized, tonic-clonic convulsions unrelated to other medical conditions in a pregnant woman with preeclampsia) [22,23].

Intertwin birthweight discordance (BWD) was calculated using the following formula: weight difference between larger twin birthweight and smaller twin birthweight divided by the larger twin birthweight. A discordant twin pair was defined as a BWD greater than 20%. Large for gestational age (LGA) was defined as birthweight above the 90th percentile for gestational age and sex, and SGA was defined as birthweight below the 10th percentile for gestational age. The birthweight percentile was assessed using the Chinese twin sex-specific standards reported by Dai et al. [24].

### 2.3. Statistical Analysis

Characteristics of study participants were compared according to the chorionicity of twin pregnancies. Neonatal outcomes were compared according to maternal GDM and/or HDP status. Continuous variables are presented as the mean (standard deviation) and categorical data are expressed as frequencies (percentages). The Shapiro-Wilk W-test was used to test the normality of data. For the difference analysis, Student’s *t* test or one-way analysis of variance was used for normally distributed continuous data and the chi-square test was used for normally distributed categorical data. The Mann–Whitney U test or Kruskal–Wallis H test was used for data that were not normally distributed. Linear and logistic regression models were used to verify the effects of HDP on neonatal outcomes in the presence of GDM or absence of GDM. Beta coefficients or odds ratios (ORs) and 95% confidence intervals (CIs) were calculated. The adjusted potential confounders included maternal age, prepregnancy BMI, nulliparity, mode of conception and chorionicity. All statistical analyses were conducted in Stata 15.0 (StataCorp, College Station, TX, USA).

## 3. Results

### 3.1. Characteristics of the Study Participants

The flow diagram for the participant selection process of this study is presented in Appendix A. A total of 1398 women with twin pregnancies were included in the final analysis, with 872 (62.4%) pregnancies being dichorionic. The comparison of the twin pregnancies based on chorionicity is presented in Table 1. Compared with dichorionic twin pregnancies, women in the monochorionic twin pregnancies group were younger (mean (SD) 29.0 (4.4) vs. 31.0 (4.2), *p* < 0.001), had a lower BMI (mean (SD) 21.2 (2.7) vs. 21.8 (3.0), *p* = 0.001), were less likely to be nulliparous (62.9% vs. 77.4%, *p* < 0.001) and delivered at an earlier gestation (mean (SD) 35.3 (2.0) vs. 36.1 (2.0), *p* < 0.001). Twin offspring delivered from monochorionic pregnancies had a lower mean birthweight (mean (SD) 2212.5 (515.6) vs. 2444.2 (488.0), *p* < 0.001) and a greater intertwin birthweight discordance (mean (SD) 14.2 (12.5) vs. 11.0 (10.2), *p* < 0.001) than those born from dichorionic pregnancies. Additionally, 400 (28.6%) women were diagnosed with GDM, and women with monochorionic twin pregnancies had a lower incidence of GDM than women with dichorionic twin pregnancies (24.9% vs. 30.9%, *p* = 0.017). A total of 128 cases were diagnosed as having HDP, and no difference was observed between the two groups: 43 (8.2%) in the monochorionic group and 85 (9.8%) in the dichorionic group (*p* = 0.323).

### 3.2. The Incidence of HDP in Twin Pregnancies Complicated by GDM

Table 2 shows that the incidence of HDP in pregnancies complicated by GDM was significantly higher than that in pregnancies not complicated by GDM (13.3% vs. 7.5%, *p* = 0.001). After adjusting for maternal age, prepregnancy BMI, nulliparity and mode of conception, the risk of HDP in women with twin pregnancies complicated by GDM was 1.57-fold higher than that in those with twin pregnancies not complicated by GDM (aOR = 1.57, 95% CI: 1.07–2.31, *p* = 0.022). However, subgroup analyses showed that GDM increased the risk of HDP only in women with MC twin pregnancies (14.5% vs. 6.1%, *p* = 0.002; aOR = 2.13, 95% CI: 1.08–4.19, *p* = 0.029). Although the incidence of HDP in pregnancies complicated by GDM was higher than that in pregnancies not complicated by GDM among DC twin pregnancies, the difference was not statistically significant (12.6% vs. 8.5%, *p* = 0.052).

### 3.3. Neonatal Outcomes According to Maternal GDM and HDP Status

To determine the effect of the co-existence of GDM and HDP on neonatal outcomes, we divided twin pregnancies into four groups according to maternal GDM and HDP status: no-GDM and no-HDP; HDP and GDM; GDM and no-HDP; and HDP and no-GDM. The results displayed that the incidence of preterm birth < 34 GA and discordant twins was highest in the HDP and no-GDM group, and this group had the lightest mean birthweight and largest intertwin birthweight discordance (Appendix A). By comparing HDP-twin pregnancies complicated by GDM with those not complicated by GDM, the results exhibited that HDP-twin pregnancies complicated by GDM had a lower risk of preterm birth < 34 GA (5.7% vs. 21.3%, *p* <0.001), heavier mean birthweight (mean (SD) 2375.3 (477.8) vs. 2244.1 (518.0), *p* = 0.004) and lesser intertwin birthweight discordance (mean (SD) 12.3 (9.4) vs. 16.2 (12.4), *p* = 0.048) (Appendix A). These differences remained significant among MC twin pregnancies, but only the incidence of preterm birth < 34 GA remained significant among DC twin pregnancies in the subgroup analyses based on chorionicity (Table 3).

### 3.4. Association between HDP-Twin Pregnancies with or without GDM and Neonatal Outcomes

Appendix A shows the multivariate regression analyses for the associations between HDP and neonatal outcomes. After adjusting for maternal age, prepregnancy BMI, nulliparity, mode of conception and chorionicity, HDP was positively correlated with risk of discordant twins irrespective of GDM status (aOR = 1.72, 95% CI: 1.02–2.91, *p* = 0.041; aOR = 2.26, 95% CI: 1.56–3.28, *p* < 0.001). Only in HDP pregnancies not complicated by GDM, HDP was negatively correlated with gestational age (aβ = −0.45, 95% CI: −0.79 to −0.11, *p* = 0.009) and mean birthweight (aβ = −127.30, 95% CI: −212.24 to −42.36, *p* = 0.003) and positively correlated with intertwin birthweight discordance (aβ = 3.14, 95% CI: 2.12–5.04, *p* < 0.001) as well as risk of preterm birth < 34 GA (aOR = 1.49, 95% CI: 1.04–2.15, *p* = 0.030).

The sensitivity analysis according to chorionicity is shown in Table 4. The strength of associations between HDP and neonatal outcomes in dichorionic twin pregnancies resembles the association between HDP and neonatal outcomes in all participants. It was unexpected to find that HDP was no longer a risk factor for poor neonatal outcomes in HDP pregnancies uncomplicated by GDM among MC twin pregnancies, even though HDP was positively correlated with gestational age and mean birthweight and negatively correlated with preterm birth < 34 GA in HDP pregnancies complicated by GDM.

With respect to SGA, HDP was a risk factor for SGA in HDP pregnancies not complicated by GDM among DC twin pregnancies (aOR = 2.03, 95% CI: 1.02–4.03, *p* = 0.044), whereas the association lost statistical significance when GDM was introduced (Table 4). We did not find an association between HDP and SGA among MC twin pregnancies.

## 4. Discussion

### 4.1. Summary of Main Results

Our study findings demonstrated that GDM was a risk factor for HDP only in monochorionic twin pregnancies after controlling for maternal age, prepregnancy BMI, nulliparity and mode of conception, but HDP was not associated with adverse neonatal outcomes in monochorionic twin pregnancies irrespective of GDM status. In dichorionic twin pregnancies, HDP was associated with a higher risk of adverse neonatal outcomes among offspring unexposed to GDM, but these associations were null in cases of the co-existence of GDM and HDP.

### 4.2. Interpretation of Study Findings and Comparison with Published Literature

Epidemiological investigations have provided evidence that GDM is associated with a higher risk of HDP in singleton pregnancies [6,25,26], but the impact of GDM on HDP risk in twin pregnancies provides conflicting results. In some studies, researchers noted that GDM increased the risk of HDP [8,10,13,27,28,29], yet other studies determined that GDM was not associated with a higher risk of HDP [6,30,31]. Despite the higher rate of preeclampsia in twin pregnancies exposed to GDM, GDM was not found to be a risk factor after adjusting for multiple variables [31]. In the current retrospective study, our findings demonstrated that the presence of GDM increased the incidence of HDP, particularly in MC twin pregnancies, which was in accordance with our previously reported results based on a longitudinal twin pregnancy cohort [15].

In twin pregnancies, advanced maternal age and prepregnancy obesity were attributed to a higher GDM and HDP risk, which is similar to high risk factors for GDM and HDP in singleton pregnancy [32,33]. However, the molecular mechanism of GDM and HDP in twin pregnancy is different from that in singleton pregnancy. It has been reported that a larger placenta area is associated with GDM in twin pregnancies [34]. The risk of preeclampsia in twin pregnancies may not be related to abnormal placentation but is mainly associated with an increase in angiogenic factors caused by an increase in placental mass [35,36]. Thus, we speculated that GDM associated with increased HDP risk may be due to hyperglycemia upregulating angiogenic factors, such as sEng and sFlt-1, which are biomarkers for preeclampsia [37], and hyperglycemia-promoted inflammation and accumulation of reactive oxidative species impair vascular cells that eventually damage vascular function [38]. With respect to such an association only found in MC twin pregnancies, one possible explanation might be the higher incidence of HDP in DC twin pregnancies masking the effect of GDM on HDP and resulting in a lack of power to find this association in this study. 

The association between HDP and neonatal outcomes in twin pregnancies was investigated. Of these reports, the associations between HDP and intertwin birthweight discordance as well as SGA or IUGR were most studied. In a small retrospective study, because the authors defined IUGR according to singleton-based birthweight percentiles, no correlation was found between preeclampsia and IUGR [39]. Subsequently, researchers pointed out that HDP was strongly associated with SGA or IUGR by using appropriate birthweight references, yet the results about the modification effect of chorionicity on this association were conflicting [40,41]. Che et al. reported that HDP was positively correlated with intertwin birthweight discordance only in DC twin neonates [19]. In addition, SGA in at least one twin but not intertwin birthweight discordance was a significant indicator for HDP irrespective of chorionicity [20,42] after adjustment, while Qiao et al. found that intertwin growth discordance is associated with an increased risk for preeclampsia only in dichorionic twin pregnancies [21]. However, almost all of these studies did not take GDM into consideration when exploring the association of HDP and SGA or intertwin birthweight discordance.

Our results revealed that HDP was a risk factor for greater intertwin birthweight discordance, lower birthweight and a higher risk of SGA only in DC twin neonates unexposed to maternal GDM. If GDM was introduced, these associations lost significance. Hyperglycemia seemed to play a protective role in attenuating the adverse impact of HDP on neonatal outcomes in DC twins. Due to marginal or velamentous cord insertion being a common cause of SGA in twins [43], hyperglycemia compensated for the insufficient nutrient supply of fetuses suspected of being SGA, thus decreasing the risk for SGA as well as lowering the intertwin weight discordance.

No associations were found between HDP and adverse neonatal outcomes in MC twin neonates unexposed to GDM, which was in accordance with findings presented by Che et al. However, MC twin neonates born to women with both GDM and HDP had a longer gestational age, a lower preterm birth risk and a heavier birthweight. These results suggested that neonatal outcomes in MC twins were more affected by GDM than by HDP, while HDP might have little effect on MC twin neonatal outcomes.

### 4.3. Clinical and Research Implications

HDP affected fetal growth and was a risk factor for iatrogenic prematurity in twin pregnancies, while GDM accelerated fetal growth. However, both the prevalence of GDM and the prevalence of HDP were much higher in twin pregnancies than in singleton pregnancies, and these two conditions may co-exist in a woman with a twin pregnancy. Evidence in the literature is very limited regarding the neonatal outcomes of twins exposed to GDM and HDP. Our findings suggested that GDM can counteract the negative effect of HDP on neonatal outcomes. The association between the co-existence of GDM and HDP and fetal growth in twins might facilitate personalized antenatal management to balance the iatrogenic prematurity risk and improve neonatal outcomes.

### 4.4. Strengths and Limitations

To our knowledge, this was the first study to identify the role of the co-existence of GDM and HDP on neonatal outcomes in twin pregnancies. The main strength was that we defined SGA according to Chinese twin sex-specific standards, since accumulating studies have revealed that twin-specific growth charts should be used to assess fetal growth to reduce the overdiagnosis of growth abnormalities [41]. In addition, we excluded twin pregnancies complicated by TTTS, TAPS and TRAPS as well as intrauterine fetal death to reduce the bias of assessment of neonatal outcomes.

Several limitations should be taken into consideration. First, we did not collect detailed information on the glucose levels of GDM pregnancies; thus, we cannot evaluate the possible range of blood glucose levels that can obviously increase the risk of HDP. Second, some outcomes were not included in the neonatal outcomes, such as neonatal respiratory distress syndrome and neonatal intensive care unit admission, since neonates at high risk were immediately transferred to the Affiliated Children Hospital of Chongqing Medical University after delivery, resulting in difficulty in collecting related data. Last, this study was a single-center study, and the findings have reference value finiteness for populations in other regions. Multicenter studies should be conducted to clarify the impact of the co-existence of GDM and HDP on neonatal outcomes.

## 5. Conclusions

This study demonstrated the impact of the co-existence of GDM and HDP on neonatal outcomes in twin pregnancies. Further studies are needed to clarify the molecular mechanism by which GDM increases the risk of HDP as well as the blood glucose control range of women with twin pregnancies complicated by HDP to optimize the antenatal management of mothers with GDM and HDP and to improve neonatal outcomes.

## Figures and Tables

**Table 1 jcm-12-01096-t001:** Characteristics of study participants.

Variables	DC (*n* = 872)	MC (*n* = 526)	*p*-Value
Age, years	31.0 ± 4.2	29.0 ± 4.4	<0.001
Prepregnancy BMI, kg/m^2^	21.8 ± 3.0	21.2 ± 2.7	0.001
Nulliparous, *n* (%)	675 (77.4)	331 (62.9)	<0.001
Assisted conception, *n* (%)	523 (60.0)	48 (9.1)	<0.001
GDM, *n* (%)	269 (30.9)	131 (24.9)	0.017
HDP, *n* (%)	85 (9.8)	43 (8.2)	0.323
Gestational weight gain, kg	17.3± 5.8	17.6 ± 5.9	0.460
Gestational age at delivery, weeks	36.1 ± 2.0	35.5 ± 2.0	<0.001
Preterm birth < 34 weeks, *n* (%)	101 (11.6)	107 (20.3)	<0.001
Cesarean delivery	850 (97.5)	509 (96.8)	0.724
Mean birthweight, g	2442.2 ± 488.0	2212.5 ± 515.6	<0.001
Larger twin birthweight, g	2587.0 ± 470.7	2375.3 ± 459.2	<0.001
Smaller twin birthweight, g	2297.4 ± 461.6	2049.8 ± 517.9	<0.001
Intertwin weight discordance, %	11.0 ± 10.2	14.2 ± 12.5	<0.001
Discordant twins, *n* (%)	123 (14.1)	141 (26.8)	<0.001

± represents SD. DC: dichorionic, MC: monochorionic, BMI: body mass index, GDM: gestational diabetes mellitus, HDP: hypertension disorders of pregnancy.

**Table 2 jcm-12-01096-t002:** HDP incidence among twin pregnancies with GDM, stratified by chorionicity.

	GDM	No-GDM	*p*-Value	OR, 95% CI	aOR, 95% CI *	*p*-Value Adjusted
Total						
HDP	53 (13.3)	75 (7.5)	0.001	1.88 (1.29, 2.73)	1.57 (1.07, 2.31)	0.022
No-HDP	347 (86.7)	923 (92.5)				
MC						
HDP	19 (14.5)	24 (6.1)	0.002	2.62 (1.39, 4.96)	2.13 (1.08, 4.19)	0.029
No-HDP	112 (85.5)	371 (93.9)				
DC						
HDP	34 (12.6)	51 (8.5)	0.052	1.57 (0.99, 2.48)	1.32 (0.82, 2.12)	0.259
No-HDP	235 (87.4)	552 (91.5)				

DC: dichorionic, MC: monochorionic, GDM: gestational diabetes mellitus, HDP: hypertension disorders of pregnancy, OR: odds ratio, CI: confidence interval. * Adjusted for maternal age, prepregnancy BMI, nulliparity and mode of conception.

**Table 3 jcm-12-01096-t003:** Neonatal outcomes among HDP-twin pregnancies complicated by GDM or not, stratified by chorionicity.

Variables	DC Twin Pregnancies			MC Twin Pregnancies		
	HDP and GDM (*n* = 68)	HDP and No-GDM (*n* = 102)	*p*-Value	HDP and GDM (*n* = 38)	HDP and No-GDM (*n* = 48)	*p*-Value
Gestational age at delivery, weeks	36.0 ± 1.7	35.7 ± 2.1	0.331	36.1 ± 1.3	34.9 ± 2.2	0.003
Preterm birth < 34 weeks, *n* (%)	4 (5.9)	20 (19.6)	0.012	2 (5.3)	12 (25.0)	0.014
Mean birthweight, g	2387.3 ± 510.2	2292.3 ± 492.6	0.226	2353.0 ± 68.2	2141.9 ± 559.9	0.045
LGA, *n* (%)	8 (11.8)	6 (5.9)	0.172	2 (5.3)	4 (8.3)	0.579
SGA, *n* (%)	7 (10.3)	11 (10.8)	0.919	5 (13.2)	8 (16.7)	0.652
Intertwin birthweight discordance, %	12.0 ± 10.3	15.2 ± 13.2	0.109	12.1 ± 8.7	18.3 ± 9.5	0.003
Discordant twins, *n* (%)	18 (26.5)	32 (31.4)	0.492	6 (15.8)	18 (37.5)	0.026

± represents SD. DC: dichorionic, MC: monochorionic, GDM: gestational diabetes mellitus, HDP: hypertension disorders of pregnancy, LGA: large for gestational age, SGA: small for gestational age.

**Table 4 jcm-12-01096-t004:** Association between HDP-twin pregnancies with or without GDM and neonatal outcomes in twin pregnancies.

Variables	HDP and GDM β/OR (95% CI) *	*p*-Value	HDP and No-GDM β/OR (95% CI) *	*p*-Value
MC twin pregnancies				
Gestational age at delivery	1.06 (0.43, 1.69)	0.001	−0.59 (−1.20, 0.02)	0.057
Preterm birth	0.44 (0.20, 0.97)	0.042	2.15 (0.98, 4.71)	0.055
Birth weight	174.90 (8.91, 340.89)	0.039	−61.36 (−217.69, 94.98)	0.441
LGA	0.73 (0.16, 3.39)	0.687	1.76 (0.56, 5.47)	0.332
SGA	1.22 (0.42, 3.50)	0.715	1.14 (0.51, 2.53)	0.754
Intertwin weight discordance	−0.02 (−0.05, 0.02)	0.438	0.03 (−0.01, 0.06)	0.105
Discordant twins	0.57 (0.22, 1.47)	0.245	1.45 (0.78, 2.69)	0.239
DC twin pregnancies				
Gestational age at delivery	−0.10 (−0.59, 0.39)	0.680	−0.39 (−0.80, 0.03)	0.066
Preterm birth	1.64 (0.93, 2.87)	0.085	1.32 (0.87, 2.01)	0.194
Birth weight	−90.64 (−213.86, 32.58)	0.149	−156.97 (−257.92, −56.02)	0.002
LGA	1.08 (0.47, 2.47)	0.864	0.55 (0.23, 1.29)	0.169
SGA	2.43 (0.93, 6.36)	0.070	2.03 (1.02, 4.03)	0.044
Intertwin weight discordance	0.02 (−0.00, 0.04)	0.082	0.04 (0.02, 0.06)	0.000
Discordant twins	3.37 (1.75, 6.49)	0.000	2.83 (1.78, 4.48)	0.000

DC: dichorionic, MC: monochorionic, GDM: gestational diabetes mellitus, HDP: hypertension disorders of pregnancy, LGA: large for gestational age, SGA: small for gestational age, OR: odds ratio, CI: confidence interval. * Adjustment for maternal age, prepregnancy BMI, nulliparity and mode of conception.

## Data Availability

The data presented in this study are available on request from the corresponding author.

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
