# Peer review of "Impact of Gestational Diabetes and Hypertension Disorders of Pregnancy on Neonatal Outcomes in Twin Pregnancies Based on Chorionicity"

_jcm, 2023, doi:10.3390/jcm12031096_

Round 1

Reviewer 1 Report

Review

Comments

Li and colleagues looked to evaluate the co-existence of GDM and HDP within twin pregnancies and the impact of chorionicity on this association. They showed HDP was more commonly present in pregnancies with GDM than without but this was only seen in MC and not DC pregnancies. They suggest these factors together attenuate the adverse impacts of HDP.

I found the manuscript interesting to read but hard to follow. I think this was for chiefly 3 reasons

1)      The message was not clear and seemed to change between a) the impact of chorionicity and b) the impact of the co-existence of GDM and HDP, the interface between the two became muddled

2)      The suggestion of causality between GDM and HDP see below

3)      The lack of adjustment for gestational age particularly comparing groups with marked difference in premature birth (GDM and HDP together had very low prem birth) significantly impacts  

Overall I think the manuscript would benefit from significant clarification and simplification .  The discussion in particular runs far ahead of the data.

I think rewritten with one simple message around “the impact on neonatal outcomes of coexistent HDP and GDM in twin pregnancies changes with chorionicty” would make for an impactful manuscript but currently the difficulty for the reader in interpreting the data loses this

Abstract

As space allows make the meaning of discordant clear

Aims

“ investigate the effects of GDM on the incidence of HDP “ this implies causality of GDM on HDP which is less likely than shared pathways, I would recommend changing, perhaps the association of GDM with HDP

Methods

A flow diagram would be helpful as appears 2 cohorts from different places (2045 and 2006) but not completely clear. Also good to see number meeting each exclusion. The dates covered for 2006 women need to be stated (even if same as 2045 which appears case from abstract).

Is screening for GDM universal in this population or based on risk of GDM?

It doesn’t look like birthweight was adjusted for gestational age, it should be especially given delivery >=28 weeks included.

Results

Where to 1398 women come from again a study flow diagram would be helpful

As above table S1 has data for >2500 women, needs clarification as to these additional women

Was GDM different between chronicity groups if adjusted for maternal age?

Line 123 the phrase “uncomplicated GDM pregnancies” is unclear, I suggest should be written as not complicated by GDM

All tables need to state +- represent SD. Table S1 is SD correct for birthweight for GDM+ HDP group? This group has only 102 women and has high % of both sga and lga (suggesting large variation in birthweight), yet smallest SD of all groups?

As above with regard to adjusting  birth weight for gestation given the significant discrepancy between groups for pre term birth in S2 S3 and Table 3 hard to know how to interpret birth weight difference.

Table S2 looks ad HDP +/- GDM could we also have broken down GDM +/- HDP ie together do they attenuate risks of GDM?

I am completely lost by table 4, I cannot understand what this shows. What comparisons are being made i.e what are OR calculated against and what do p values relate to? Main test states “resembles that of overall participant” but far more clarity is needed to explain what this actually means and the analysis undertaken. The outcome labels look like they have been carried forward from other tables as % n etc seem irrelevant given OR stated.

Discussion

“GDM was a risk factor for HDP”  as per aims this infers causation. This oversteps the data, we can say they occur together (i.e may have common risk (likely)) but not that one causes the other

“but HDP was not a risk factor” as above re causation. I think discuss in terms of association instead.

Lines 227 to 235 are very hard to follow and strike this reader as including many leaps of faith away from data.

Main conclusion does not mention twins. “This study demonstrated the impact of the co-existence of GDM and HDP on neonatal outcomes.”

Minor

In addition start of third paragraph and additionally line 135 make it harder to read.

Line 166 By not BY

Reviewer 2 Report

Li and Colleagues investigated the impact of the co-existence of gestational diabetes (GDM) 16 and hypertension disorders of pregnancy (HDP) on neonatal outcomes in twin pregnancies based 17 on chorionicity. Is a very important theme to know the coexistence of these two disorders in neonate’s outcome. Additionally, the study has a very strong methodological aspect particularly sample size of 1398. For that reason, I think the purpose of this research is meaningful.

Based on the results of this study, the authors conclude that GDM leads to an increased risk of HDP only in MC twin pregnancies, but GDM seemed to attenuate the adverse effects of HDP on perinatal outcomes in both MC and DC twin pregnancies. I generally agree with their conclusions and recommendation, however, few minor comments are as below. Additionally a retrospective ethics approval is suggested.

Introduction:

The exiting literature is poorly explored and summed up, please provide a gap in literature and thus providing a strong rationale for the study.

Methods:

Line 76-77: It is not clear why ethic approval not required? I believe all human research, even retrospective study do require ethic approval may be with individual consent waiver particularly for this kind of study.

Was Early GDM considered?

Line 117: what were nonparametric test used in the analysis?

Result and Conclusion:

Well drafted.

Author Response

Response to the reviewers’ comments

Dear Editor,

My coauthors join me in expressing our sincere appreciation for your time and effort with regard to handling this manuscript and to the reviewers for their constructive and thoughtful comments, which have strengthened the manuscript. We have critically reviewed the comments and revised the manuscript as per the reviewers’ suggestions. Our point-by-point responses to the comments are addressed below, and all changes in the revised manuscript are highlighted in yellow. We hope that the revised manuscript is satisfactory to you and the reviewers and can be considered for publication in your prestigious journal.

Comments

Reviewer 2

Li and Colleagues investigated the impact of the co-existence of gestational diabetes (GDM) and hypertension disorders of pregnancy (HDP) on neonatal outcomes in twin pregnancies based on chorionicity. Is a very important theme to know the coexistence of these two disorders in neonate’s outcome. Additionally, the study has a very strong methodological aspect particularly sample size of 1398. For that reason, I think the purpose of this research is meaningful.

Based on the results of this study, the authors conclude that GDM leads to an increased risk of HDP only in MC twin pregnancies, but GDM seemed to attenuate the adverse effects of HDP on perinatal outcomes in both MC and DC twin pregnancies. I generally agree with their conclusions and recommendation, however, few minor comments are as below. Additionally, a retrospective ethics approval is suggested.

Response: Thank you for your suggestion on the ethics approval. We will apply an ethics approval for this retrospective study, but we cannot get the ethics approval within the deadline of Submit Revised Manuscript.

Introduction:

The exiting literature is poorly explored and summed up, please provide a gap in literature and thus providing a strong rationale for the study.

Response: We revised the third paragraph in the Introduction section to provide a stronger rationale for the current study in the revised version.

Methods:

Line 76-77: It is not clear why ethic approval not required? I believe all human research, even retrospective study do require ethic approval may be with individual consent waiver particularly for this kind of study.

Response: We are sorry that we did not require ethical committee approval, but we informed the twin pregnant women that their medical data would be used for scientific research and obtained oral agreement from the participants.

Was Early GDM considered?

Response: Because only fasting glucose level was tested in early pregnancy and GDM screening is routinely performed during 23 and 28 gestational weeks in twin pregnancies, we did not consider early GDM in the current study.

Line 117: what were nonparametric test used in the analysis?

Response: The Mann-Whitney U test was used for comparisons between two groups, and the Kruskal-Wallis H test was used for comparisons between four groups. We have added the detailed nonparametric test methods to the Statistical analysis paragraph in the revised version.

Result and Conclusion:

Well drafted.

Round 2

Reviewer 2 Report

All the comments have been resolved.

Author Response

We are greatful to reviewer 2 for his/her effort reviewing our paper and his/her positive feedback.